# Learning Language-guided Adaptive Hyper-modality Representation for Multimodal Sentiment Analysis

**Haoyu Zhang**[1,2],**Yu Wang**[2],**Guanghao Yin**[2],**Kejun Liu**[2],**Yuanyuan Liu**[2,3],**Tianshu Yu**[1,4*]

[1]The Chinese University of Hong Kong, Shenzhen
[2]China University of Geosciences, Wuhan
[3]Nanyang Technological University
[4] Shenzhen Institute of Artificial Intelligence and Robotics for Society
[1,4]{zhanghaoyu, yutianshu}@cuhk.edu.cn
[2]{zhanghaoyu, vvy190701, ygh2, liukejun, liuyy}@cug.edu.cn
[3]scse-yyliu@ntu.edu.sg

## Abstract

Though Multimodal Sentiment Analysis (MSA) proves effective by utilizing rich information from multiple sources (*e.g.,* language, video, and audio), the potential sentiment-irrelevant and conflicting information across modalities may hinder the performance from being further improved. To alleviate this, we present Adaptive Language-guided Multimodal Transformer (ALMT), which incorporates an Adaptive Hyper-modality Learning (AHL) module to learn an irrelevance/conflict-suppressing representation from visual and audio features under the guidance of language features at different scales. With the obtained hyper-modality representation, the model can obtain a complementary and joint representation through multimodal fusion for effective MSA. In practice, ALMT achieves state-of-the-art performance on several popular datasets (*e.g.,* MOSI, MOSEI and CH-SIMS) and an abundance of ablation demonstrates the validity and necessity of our irrelevance/conflict suppression mechanism.

## 1 Introduction

Multimodal Sentiment Analysis (MSA) focuses on recognizing the sentiment attitude of humans from various types of data, such as video, audio, and language. It plays a central role in several applications, such as healthcare and human-computer interaction (Jiang et al., 2020; Qian et al., 2019). Compared with unimodal methods, MSA methods are generally more robust by exploiting and exploring the relationships between different modalities, showing significant advantages in improving the understanding of human sentiment.

Most recent MSA methods can be grouped into two categories: representation learning-centered methods (Hazarika et al., 2020; Yang et al., 2022;

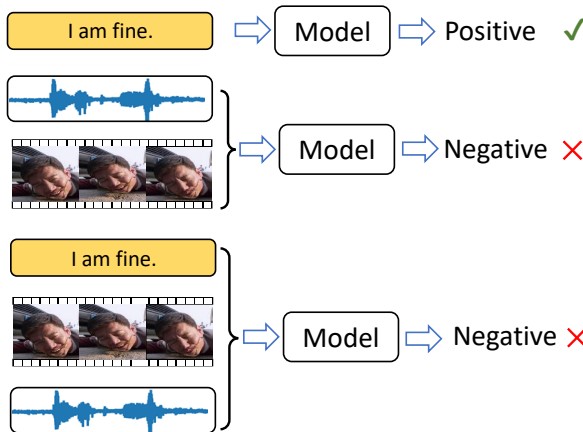

Figure 1: In multimodal sentiment analysis, language modality usually is a dominant modality in all modalities, while audio and visual modalities not contributing as much to performance as language modality.

Yu et al., 2021; Han et al., 2021; Guo et al., 2022) and multimodal fusion-centered methods (Zadeh et al., 2017; Liu et al., 2018; Tsai et al., 2019a; Huang et al., 2020). The representation learning-centered methods mainly focus on learning refined modality semantics that contains rich and varied human sentiment clues, which can further improve the efficiency of multimodal fusion for relationship modelling. On the other hand, the multimodal fusion-centered methods mainly focus on directly designing sophisticated fusion mechanisms to obtain a joint representation of multimodal data. In addition, some works and corresponding ablation studies (Hazarika et al., 2020; Rahman et al., 2020; Guo et al., 2022) further imply that various modalities contribute differently to recognition, where language modality stands out as the dominant one. We note, however, information from different modalities may be ambiguous and conflicting due to sentiment-irrelevance, especially from non-dominating modalities (e.g., lighting and head pose in video and background noise in audio). Such

---
*Corresponding author

disruptive information can greatly limit the performance of MSA methods. We have observed this phenomenon in several datasets (see Section 4.5.1) and an illustration is in Figure 1. To the best of our knowledge, there has never been prior work explicitly and actively taking this factor into account.

Motivated by the above observation, we propose a novel Adaptive Language-guided Multimodal Transformer (ALMT) to improve the performance of MSA by addressing the adverse effects of disruptive information in visual and audio modalities. In ALMT, each modality is first transformed into a unified form by using a Transformer with initialized tokens. This operation not only suppresses the redundant information across modalities, but also compresses the length of long sequences to facilitate efficient model computation. Then, we introduce an Adaptive Hyper-modality Learning (AHL) module that uses different scales of language features with dominance to guide the visual and audio modalities to produce the intermediate hyper-modality token, which contains less sentiment-irrelevant information. Finally, we apply a cross-modality fusion Transformer with language features serving as query and hyper-modality features serving as key and value. In this sense, the complementary relations between language and visual and audio modalities are implicitly reasoned, achieving robust and accurate sentiment predictions. In summary, the major contributions of our work can be summarized as:

- We present a novel multimodal sentiment analysis method, namely Adaptive Language-guided Multimodal Transformer (ALMT), which for the first time explicitly tackles the adverse effects of redundant and conflicting information in auxiliary modalities (*i.e.*, visual and audio modalities), achieving a more robust sentiment understanding performance.

- We devise a novel Adaptive Hyper-modality Learning (AHL) module for representation learning. The AHL uses different scales of language features to guide the visual and audio modalities to form a hyper modality that complements the language modality.

- ALMT achieves state-of-the-art performance in several public and widely adopted datasets. We further provide in-depth analysis with rich empirical results to demonstrate the validity and necessity of the proposed approach.

## 2 Related Work

In this part, we briefly review previous work from two perspectives: multimodal sentiment analysis and Transformers.

### 2.1 Multimodal Sentiment Analysis

As mentioned in the section above, most previous MSA methods are mainly classified into two categories: representation learning-centered methods and multimodal fusion-centered methods.

For representation learning-centered methods, Hazarika et al. (2020) and Yang et al. (2022) argued representation learning of multiple modalities as a domain adaptation task. They respectively used metric learning and adversarial learning to learn the modality-invariant and modality-specific subspaces for multimodal fusion, achieving advanced performance in several popular datasets. Han et al. (2021) proposed a framework named MMIM that improves multimodal fusion with hierarchical mutual information maximization. Rahman et al. (2020) and Guo et al. (2022) devised different architectures to enhance language representation by incorporating multimodal interactions between language and non-verbal behavior information. However, these methods do not pay enough attention to sentiment-irrelevant redundant information that is more likely to be present in visual and audio modalities, which limits the performance of MSA.

For multimodal fusion-centered methods, Zadeh et al. (2017) proposed a fusion method (TFN) using a tensor fusion network to model the relationships between different modalities by computing the cartesian product. Tsai et al. (2019a) and Huang et al. (2020) introduced a multimodal Transformer to align the sequences and model long-range dependencies between elements across modalities. However, these methods directly fuse information from uni-modalities, which is more accessible to the introduction of sentiment-irrelevant information, thus obtaining sub-optimal results.

### 2.2 Transformer

Transformer is an attention-based building block for machine translation introduced by Vaswani et al. (2017). It learns the relationships between tokens by aggregating data from the entire sequence, showing an excellent modeling ability in various tasks, such as natural language processing, speech processing, and computer vision, etc. (Kenton and

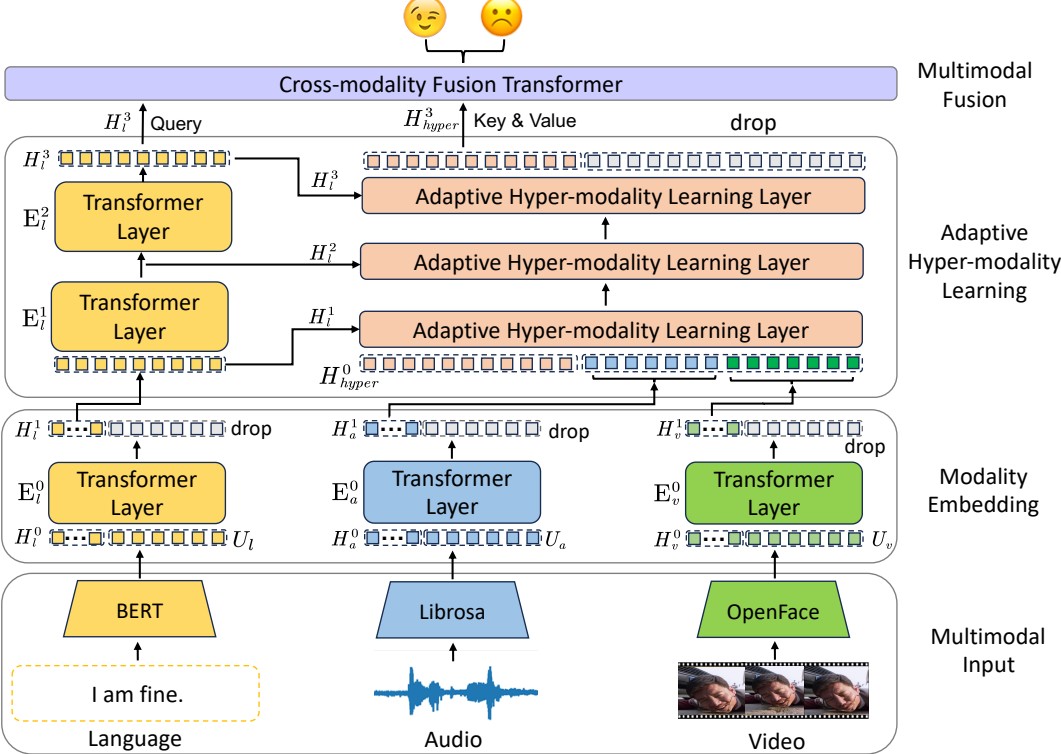

Figure 2: Processing pipeline of the proposed ALMT for multimodal sentiment analysis (MSA). With the multimodal input, we first apply three Transformer layers to embed modality features with low redundancy. Then, we employ a Hyper-modality Learning (AHL) module to learn a hyper-modality representation from visual and audio modalities under the guidance of language features at different scales. Finally, a Cross-modality Fusion Transformer is applied to incorporate hyper-modality features based on their relations to the language features, thus obtaining a complementary and joint representation for MSA.

Toutanova, 2019; Carion et al., 2020; Chen et al., 2022; Liu et al., 2023a). In MSA, this technique has been widely used for feature extraction, representation learning, and multimodal fusion (Tsai et al., 2019a; Huang et al., 2020; Liu et al., 2023b; Yuan et al., 2021).

## 3 Method

### 3.1 Overview

The overall processing pipeline of the proposed Adaptive Language-guided Multimodal Transformer (ALMT) for robust multimodal sentiment analysis is in Figure 2. As shown, ALMT first extracts unified modality features from the input. Then, Adaptive Hyper-Modality Learning (AHL) module is employed to learn the adaptive hyper-modality representation with the guidance of language features at different scales. Finally, we apply a Cross-modality Fusion Transformer to synthesize the hyper-modality features with language features as anchors, thus obtaining a language-guided hyper-modality network for MSA.

### 3.2 Multimodal Input

Regarding the multimodal input, each sample consists of language ($l$), audio ($a$), and visual ($v$) sources. Referring to previous works, we first obtain pre-computed sequences calculated by BERT (Kenton and Toutanova, 2019), Librosa (McFee et al., 2015), and OpenFace (Baltrusaitis et al., 2018), respectively. Then, we denote these sequence inputs as $U_m \in \mathbb{R}^{T_m \times d_m}$, where $m \in \{l, v, a\}$, $T_m$ is the sequence length and $d_m$ is the vector dimension of each modality. In practice, $T_m$ and $d_m$ are different on different datasets. For example, on the MOSI dataset, $T_v, T_a, T_l, d_a, d_v$ and $d_l$ are 50, 50, 50, 5, 20, and 768, respectively.

### 3.3 Modality Embedding

With multimodal input $U_m$, we introduce three Transformer layers to unify features of each modality, respectively. More specifically, we randomly initialize a low-dimensional token $H_m^0 \in \mathbb{R}^{T \times d_m}$ for each modality and use the Transformer to embed the essential modality information to these to-

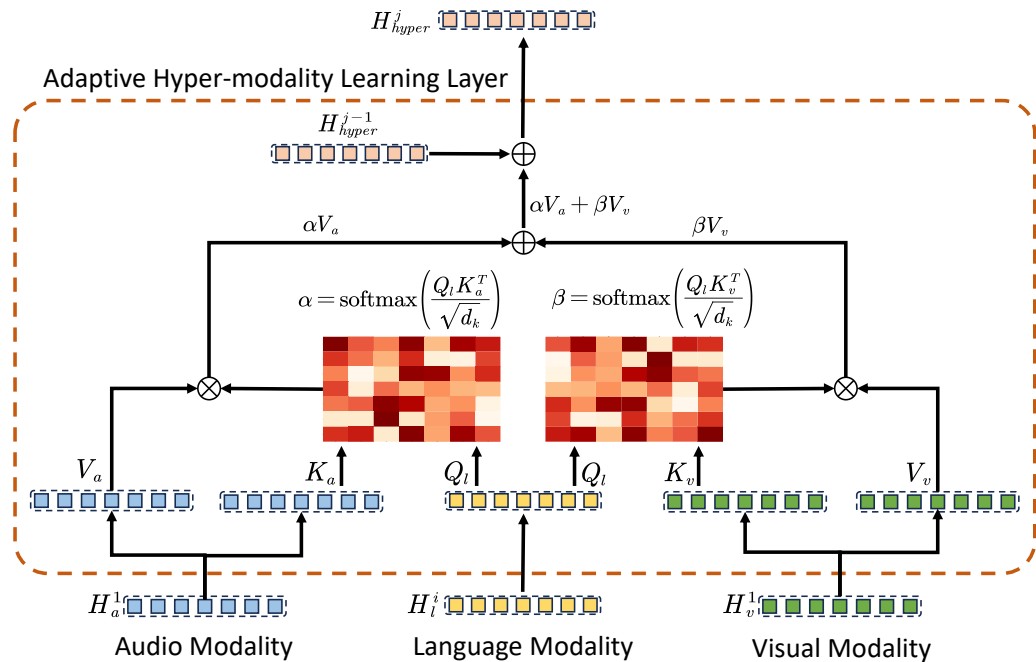

Figure 3: An example of the Adaptive Hyper-modality Learning (AHL) Layer.

kens :

$$H_m^1 = \mathrm{E}_m^0(\mathrm{concat}(H_m^0, U_m), \theta_{E_m^0}) \in \mathbb{R}^{T \times d} \quad (1)$$

where $H_m^1$ is the unified feature of each modality $m$ with a size of $T \times d$, $\mathrm{E}_m^0$ and $\theta_{E_m^0}$ respectively represent the modality feature extractor and corresponding parameters, $\mathrm{concat}(\cdot)$ represent the concatenation operation.

In practice, $T$ and $d$ are set to 8 and 128, respectively. The structure of the transformer layer is designed as the same as the Vision Transformer (VIT) (Dosovitskiy et al., 2021) with a depth setting of 1. Moreover, it is worth noting that transferring the essential modality information to initialized low-dimensional tokens is beneficial to decrease the redundant information that is irrelevant to human sentiment, thus achieving higher efficiency with lesser parameters.

## 3.4 Adaptive Hyper-modality Learning

After modality embedding, we further employ an Adaptive Hyper-modality Learning (AHL) module to learn a refined hyper-modality representation that contains relevance/conflict-suppressing information and highly complements language features. The AHL module consists of two Transformer layers and three AHL layers, which aim to learn language features at different scales and adaptively learn a hyper-modality feature from visual and audio modalities under the guidance of

language features. In practice, we found that the language features significantly impact the modeling of hyper-modality (with more details in section 4.5.4).

### 3.4.1 Construction of Two-scale Language Features

We define the feature $H_l^1$ as low-scale language feature. With the feature, we introduce two Transformer layers to learn language features at middle-scale and high-scale (*i.e.* $H_l^2$ and $H_l^3$). Different from the Transformer layer in the modality embedding stage that transfers essential information to an initialized token, layers in this stage directly model the language features:

$$H_l^i = \mathrm{E}_l^i(H_l^{i-1}, \theta_{E_l^i}) \in \mathbb{R}^{T \times d} \quad (2)$$

where $i \in \{2, 3\}$, $H_l^i$ is language features at different scales with a size of $T \times d$, $\mathrm{E}_l^i$ and $\theta_{E_l^i}$ represents the $i$-th Transformer layer for language features learning and corresponding parameters. In practice, we used 8-head attention to model the information of each modality.

### 3.4.2 Adaptive Hyper-modality Learning Layer

With the language features of different scales $H_l^i$, we first initialize a hyper-modality feature $H_{hyper}^0 \in \mathbb{R}^{T \times d}$, then update $H_{hyper}^0$ by calculating the relationship between obtained language

features and two remaining modalities using multi-head attention (Vaswani et al., 2017). As shown in Figure 3, using the extracted $H_l^i$ as query and $H_a^1$ as key, we can obtain the similarity matrix $\alpha$ between language features and audio features :

$$
\begin{aligned}
\alpha &= \text{softmax}(\frac{Q_l K_a^T}{\sqrt{d_k}}) \\
&= \text{softmax}(\frac{H_l^i W_{Q_l} W_{K_a}^T H_a^{1T}}{\sqrt{d_k}}) \in \mathbb{R}^{T \times T}
\end{aligned}
\tag{3}
$$

where softmax represents weight normalization operation, $W_{Q_l} \in \mathbb{R}^{d \times d_k}$ and $W_{K_a} \in \mathbb{R}^{d \times d_k}$ are learnable parameters, $d_k$ is the dimension of each attention head. In practice, we used 8-head attention and set $d_k$ to 16.

Similar to $\alpha$, $\beta$ represents the similarity matrix between language modality and visual modality:

$$
\begin{aligned}
\beta &= \text{softmax}(\frac{Q_l K_v^T}{\sqrt{d_k}}) \\
&= \text{softmax}(\frac{H_l^i W_{Q_l} W_{K_v}^T H_v^{1T}}{\sqrt{d_k}}) \in \mathbb{R}^{T \times T}
\end{aligned}
\tag{4}
$$

where $W_{K_v} \in \mathbb{R}^{d \times d_k}$ is learnable.

Then the hyper-modality features $H_{hyper}^j$ can be updated by weighted audio features and weighted visual features as:

$$
\begin{aligned}
H_{hyper}^j &= H_{hyper}^{j-1} + \alpha V_a + \beta V_v \\
&= H_{hyper}^{j-1} + \alpha H_a^1 W_{V_a} + \beta H_v^1 W_{V_v}
\end{aligned}
\tag{5}
$$

where $j \in \{1, 2, 3\}$ and $H_{hyper}^j \in \mathbb{R}^{T \times d}$ respectively represent the $j$-th AHL layer and corresponding output hyper-modality features, $W_{V_a} \in \mathbb{R}^{d \times d_k}$ and $W_{V_v} \in \mathbb{R}^{d \times d_k}$ are learnable parameters.

### 3.5 Multimodal Fusion and Output

In the Multimodal Fusion, we first obtain a new language feature $H_l$ and $H_{hyper}$ and a new hyper-modality feature by respectively concatenating initialized a token $H_0 \in \mathbb{R}^{1 \times d}$ with $H_{hyper}^3$ and $H_l^3$. Then we apply Cross-modality Fusion Transformer to transfer the essential joint and complementary information to these tokens. In practice, the Cross-modality Fusion Transformer fuse the language features $H_l$ (serving as the query) and hyper-modality features $H_{hyper}$ (serving as the key and value), thus obtaining a joint multimodal representation $H \in \mathbb{R}^{1 \times d}$ for final sentiment analysis. We denote the Cross-modality Fusion Transformer as

CrossTrans, so the fusion process can be written as:

$$
H_l = \text{Concat}(H_0, H_l^3) \in \mathbb{R}^{(T+1) \times d} \tag{6}
$$

$$
H_{hyper} = \text{Concat}(H_0, H_{hyper}^3) \in \mathbb{R}^{(T+1) \times d} \tag{7}
$$

$$
H = \text{CrossTrans}(H_l, H_{hyper}) \in \mathbb{R}^{1 \times d} \tag{8}
$$

After the multimodal fusion, we obtain the final sentiment analysis output $\hat{y}$ by applying a classifier on the outputs of Cross-modality Fusion Transformer $H$. In practice, we also used 8-head attention to model the relationships between language modality and hyper-modality. For more details of the Cross-modality Fusion Transformer, we refer readers to Tsai et al. (2019a).

### 3.6 Overall Learning Objectives

To summarize, our method only involves one learning objective, *i.e.,* the sentiment analysis learning loss $\mathcal{L}$, which is:

$$
\mathcal{L} = \frac{1}{N_b} \sum_{n=0}^{N_b} \|y^n - \hat{y}^n\|_2^2 \tag{9}
$$

where $N_b$ is the number of samples in the training set, $y^n$ is the sentiment label of the $n$-th sample. $\hat{y}^n$ is the prediction of our ALMT.

In addition, thanks to our simple optimization goal, compared with advanced methods (Hazarika et al., 2020; Yu et al., 2021) with multiple optimization goals, ALMT is much easier to train without tuning extra hyper-parameters. More details are shown in section 4.5.10.

## 4 Experiments

### 4.1 Datasets

We conducted extensive experiments on three popular trimodal datasets (*i.e.*, MOSI (Zadeh et al., 2016), MOSEI (Zadeh et al., 2018), and CH-SIMS (Yu et al., 2020)).

**MOSI**. The dataset comprises 2,199 multimodal samples encompassing visual, audio, and language modalities. Specifically, the training set consists of 1,284 samples, the validation set contains 229 samples, and the test set encompasses 686 samples. Each individual sample is assigned a sentiment score ranging from -3 (indicating strongly negative) to 3 (indicating strongly positive).

**MOSEI**. The dataset comprises 22,856 video clips collected from YouTube with a diverse factors (e.g., spontaneous expressions, head poses,

Table 1: Comparison on MOSI and MOSEI. Note: the best result is highlighted in bold; * represents the result is from Hazarika et al. (2020); † represents the result is from Mao et al. (2022) and its corresponding GitHub page[1].

| Method | MOSI | | | | | | MOSEI | | | | | |
| --- | --- | --- | --- | --- | --- | --- | --- | --- | --- | --- | --- | --- |
| | Acc-7 | Acc-5 | Acc-2 | F1 | MAE | Corr | Acc-7 | Acc-5 | Acc-2 | F1 | MAE | Corr |
| TFN* | 34.9 | - | -/80.8 | -/80.7 | 0.901 | 0.698 | 50.2 | - | -/82.5 | -/82.1 | 0.593 | 0.700 |
| LMF* | 33.2 | - | -/82.5 | -/82.4 | 0.917 | 0.695 | 48.0 | - | -/82.0 | -/82.1 | 0.623 | 0.677 |
| MFM* | 35.4 | - | -/81.7 | -/81.6 | 0.877 | 0.706 | 51.3 | - | -/84.4 | -/84.3 | 0.568 | 0.717 |
| MulT | 40.0 | - | -/83.0 | -/82.8 | 0.871 | 0.698 | 51.8 | - | -/82.5 | -/82.3 | 0.580 | 0.703 |
| MISA | 42.3 | - | 81.8/83.4 | 81.7/83.6 | 0.783 | 0.761 | 52.2 | - | 83.6/85.5 | 83.8/85.3 | 0.555 | 0.756 |
| PMR | 40.6 | - | -/83.6 | -/83.4 | - | - | 52.5 | - | -/83.3 | -/82.6 | - | - |
| MAG-BERT | 43.62 | - | 82.37/84.43 | 82.50/84.61 | 0.727 | 0.781 | 52.67 | - | 82.51/84.82 | 82.77/84.71 | 0.543 | 0.755 |
| Self-MM | 45.79 | - | 82.54/84.77 | 83.68/84.91 | 0.712 | 0.795 | 53.46 | - | 82.68/84.96 | 82.95/84.93 | 0.529 | 0.767 |
| MMIM | 46.65 | - | 84.14/86.06 | 84.00/85.98 | 0.700 | 0.800 | 54.24 | - | 82.24/85.97 | 82.66/85.94 | 0.526 | 0.772 |
| FDMER | 44.1 | - | -/84.6 | -/84.7 | 0.724 | 0.788 | 54.1 | - | -/86.1 | -/85.8 | 0.536 | 0.773 |
| CHFN | 48.6 | - | 84.3/86.4 | 84.2/86.2 | 0.689 | **0.809** | 54.3 | - | 83.7/86.2 | 83.9/86.1 | **0.525** | 0.778 |
| MulT† | - | 42.68 | -/- | -/- | - | - | - | 54.18 | -/- | -/- | - | - |
| MISA† | - | 47.08 | -/- | -/- | - | - | - | 53.63 | -/- | -/- | - | - |
| Self-MM† | - | 53.47 | -/- | -/- | - | - | - | 55.53 | -/- | -/- | - | - |
| **ALMT** | **49.42** | **56.41** | **84.55/86.43** | **84.57/86.47** | **0.683** | 0.805 | **54.28** | **55.96** | **84.78/86.79** | **85.19/86.86** | 0.526 | **0.779** |

Table 2: Comparison results on CH-SIMS. Note: the best result is highlighted in bold; † represents the result is from Mao et al. (2022) and its corresponding GitHub page[1].

| Method | Acc-5 | Acc-3 | Acc-2 | F1 | MAE | Corr |
| --- | --- | --- | --- | --- | --- | --- |
| TFN† | 39.30 | 65.12 | 78.38 | 78.62 | 0.432 | 0.591 |
| LMF† | 40.53 | 64.68 | 77.77 | 77.88 | 0.441 | 0.576 |
| MFM† | - | - | 75.06 | 75.58 | 0.477 | 0.525 |
| MuLT† | 37.94 | 64.77 | 78.56 | 79.66 | 0.453 | 0.564 |
| MISA† | - | - | 76.54 | 76.59 | 0.447 | 0.563 |
| MAG-BERT† | - | - | 74.44 | 71.75 | 0.492 | 0.399 |
| Self-MM† | 41.53 | 65.47 | 80.04 | 80.44 | 0.425 | 0.595 |
| **ALMT** | **45.73** | **68.93** | **81.19** | **81.57** | **0.404** | **0.619** |

occlusions, illuminations). This dataset has been categorized into 16,326 training instances, 1,871 validation instances, and 4,659 test instances. Each instance is meticulously labeled with a sentiment score ranging from -3 to 3. And the sentiment scores from -3 to 3 indicate most negative to most positive.

**CH-SIMS**. It is a Chinese multimodal sentiment dataset that comprises 2,281 video clips collected from variuous sources, such as different movies and TV serials with spontaneous expressions, various head poses, etc. It is divided into 1,368 training samples, 456 validation samples, and 457 test samples. Each sample is manually annotated with a sentiment score from -1 (strongly negative) to 1 (strongly positive).

## 4.2 Evaluation Criteria

Following prior works (Yu et al., 2020), we used several evaluation metrics, *i.e.*, binary classification accuracy (Acc-2), F1, three classification accuracy (Acc-3), five classification accuracy (Acc-5), seven classification accuracy (Acc-7), mean absolute error (MAE), and the correlation of the model's prediction with human (Corr). Moreover, on MOSI and MOSEI, agreeing with prior works (Hazarika et al., 2020), we calculated Acc-2 and F1 in two ways: negative/non-negative and negative/positive on MOSI and MOSEI datasets, respectively.

## 4.3 Baselines

To comprehensively validate the performance of our ALMT, we make a fair comparison with the several advanced and state-of-the-art methods, *i.e.*, TFN (Zadeh et al., 2017), LMF (Liu et al., 2018), MFM (Tsai et al., 2019b), MuLT (Tsai et al., 2019a), MISA (Hazarika et al., 2020), PMR (Lv et al., 2021), MAG-BERT (Rahman et al., 2020), Self-MM (Yu et al., 2021), MMIM (Han et al., 2021), FDMER (Yang et al., 2022) and CHFN (Guo et al., 2022).

## 4.4 Performance Comparison

Table 1 and Table 2 list the comparison results of our proposed method and state-of-the-art methods on the MOSI, MOSEI, and CH-SIMS, respectively.

As shown in the Table 1, the proposed ALMT obtained state-of-the-art performance in almost all metrics. On the task of more difficult and fine-grained sentiment classification (Acc-7), our model achieves remarkable improvements. For example, on the MOSI dataset, ALMT achieved a relative improvement of 1.69% compared to the second-best result obtained by CHFN. It demonstrates that eliminating the redundancy of auxiliary modalities is essential for effective MSA.

Moreover, it is worth noting that the scenarios in

[1]https://github.com/thuiar/MMSA/blob/master/results/result-stat.md

SIMS are more complex than MOSI and MOSEI. Therefore, it is more challenging to model the multimodal data. However, as shown in the Table 2, ALMT achieved state-of-the-art performance in all metrics compared to the sub-optimal approach. For example, compared to Self-MM, it achieved relative improvements with 1.44% on Acc-2 and 1.40% on the corresponding F1, respectively. Achieving such superior performance on SIMS with more complex scenarios demonstrates ALMT's ability to extract effective sentiment information from various scenarios.

### 4.5 Ablation Study and Analysis

### 4.5.1 Effects of Different Modalities

To better understand the influence of each modality in the proposed ALMT, Table 3 reports the ablation results of the subtraction of each modality to the ALMT on the MOSI and CH-SIMS datasets, respectively. It is shown that, if the AHL is removed based on the subtraction of each modality, the performance decreases significantly in all metrics. This phenomenon demonstrates that AHL is beneficial in reducing the sentiment-irrelevant redundancy of visual and audio modalities, thus improving the robustness of MSA.

In addition, we note that after removing the video and audio inputs, the performance of ALMT remains relatively high. Therefore, in the MSA task, we argue that eliminating the sentiment-irrelevant information that appears in auxiliary modalities (*i.e.,* visual and audio modalities) and improving the contribution of auxiliary modalities in performance should be paid more attention to.

Table 3: Effects of different modalities. Note: the best result is highlighted in bold.

| Method | MOSI | | CH-SIMS | |
|---|---|---|---|---|
| | Acc-7 | MAE | Acc-5 | MAE |
| **ALMT** | **49.42** | **0.683** | **45.73** | 0.404 |
| w/o Audio | 48.69 | 0.705 | 45.08 | 0.416 |
| w/o Video | 47.96 | 0.704 | 44.64 | **0.403** |
| w/o Audio & AHL | 46.91 | 0.724 | 43.54 | 0.407 |
| w/o Video & AHL | 47.08 | 0.726 | 43.76 | 0.406 |
| w/o Video & Audio | 46.79 | 0.752 | 40.26 | 0.405 |

### 4.5.2 Effects of Different Components

To verify the effectiveness of each component of our ALMT, in Table 4, we present the ablation result of the subtraction of each component on the MOSI and CH-SIMS datasets, respectively. We observe that deactivating the AHL (replaced with

feature concatenation) greatly decreases the performance, demonstrating the language-guided hyper-modality representation learning strategy is effective. Moreover, after the removal of the fusion Transformer and Modality Embedding, the performance drops again, also supporting that the fusion Transformer and Modality embedding can effectively improve the ALMT's ability to explore the sentiment information in each modality.

Table 4: Effects of different components. Note: the best result is highlighted in bold.

| Method | MOSI | | CH-SIMS | |
|---|---|---|---|---|
| | Acc-7 | MAE | Acc-5 | MAE |
| **ALMT** | **49.42** | **0.683** | **45.73** | **0.404** |
| w/o AHL | 34.40 | 0.952 | 38.29 | 0.444 |
| w/o Fusion Transformer | 48.69 | 0.703 | 43.76 | 0.410 |
| w/o Modality Embedding | 47.96 | 0.701 | 43.11 | 0.429 |

### 4.5.3 Effects of Different Query, Key, and Value Settings in Fusion Transformer

Table 5 presents the experimental results of different query, key, and value settings in Transformer on the MOSI and MOSEI datasets, respectively. We observed that ALMT can obtain better performance when aligning hyper-modality features to language features (*i.e.,* using $H_l^3$ as query and using $H_{hyper}^3$ as key and value). We attribute this phenomenon to the fact that language information is relatively clean and can provide more sentiment-relevant information for effective MSA.

Table 5: Effect of different Query, Key, and Value settings in Fusion Transformer.

| Q | K & V | MOSI | | CH-SIMS | |
|---|---|---|---|---|---|
| | | Acc-7 | MAE | Acc-5 | MAE |
| $H_{hyper}^3$ | $H_l^3$ | 48.10 | 0.707 | 44.64 | 0.410 |
| $H_l^3$ | $H_{hyper}^3$ | **49.42** | **0.683** | **45.73** | **0.404** |

### 4.5.4 Effects of the Guidance of Different Language Features in AHL

To discuss the effect of the guidance of different language features in AHL, we show the ablation result of different guidance settings on MOSI and CH-SIMS in Table 6. In practice, we replace the AHL layer that do not require language guidance with MLP layer. Obviously, we can see that the ALMT can obtain the best performance when all scals of language features (*i.e.,* $H_l^1$, $H_l^2$, $H_l^3$) involve the guidance of hyper-modality learning.

In addition, we found that the model is more difficult to converge when AHL is removed. It indicates that sentiment-irrelevant and conflicting information visual and audio modalities may limit the improvement of the model.

Table 6: Effects of different guidance of different language features in AHL. Note: the best result is highlighted in bold.

| $H_l^1$ | $H_l^2$ | $H_l^3$ | MOSI | | CH-SIMS | |
|---|---|---|---|---|---|---|
| | | | Acc-7 | MAE | Acc-5 | MAE |
| | | | 34.40 | 0.952 | 38.29 | 0.444 |
| ✓ | | | 47.38 | 0.704 | 43.54 | 0.412 |
| | ✓ | | 48.10 | 0.709 | 43.11 | 0.415 |
| | | ✓ | 48.54 | 0.711 | 43.98 | 0.412 |
| ✓ | ✓ | | 46.36 | 0.736 | 45.51 | 0.417 |
| ✓ | | ✓ | 48.10 | 0.707 | 44.20 | 0.409 |
| | ✓ | ✓ | 47.81 | 0.729 | 43.76 | 0.416 |
| ✓ | ✓ | ✓ | **49.42** | **0.683** | **45.73** | **0.404** |

### 4.5.5 Effects of Different Fusion Techniques

To analyze the effects of different fusion techniques, we conducted some experiments, whose results are shown in the table 7. Obviously, on the MOSI dataset, the use of our Cross-modality Fusion Transformer to fuse language features and hyper-modality features is the most effective. On the CH-SIMS dataset, although TFN achieves better performance on the MAE metric, its Acc-5 is lower. Overall, using Transformer for feature fusion is an effective way.

Table 7: Effects of different fusion techniques.

| Fusion Technique | MOSI | | CH-SIMS | |
|---|---|---|---|---|
| | Acc-7 | MAE | Acc-5 | MAE |
| Concatenation | 48.69 | 0.703 | 43.76 | 0.410 |
| Addition | 46.36 | 0.706 | 42.45 | 0.411 |
| GRU | 47.81 | 0.710 | 44.86 | 0.414 |
| Tensor Fusion (TFN) | 47.23 | 0.710 | 44.20 | **0.403** |
| Low-rank Fusion (LMF) | 46.65 | 0.715 | 45.08 | 0.408 |
| Ours | **49.42** | **0.683** | **45.73** | 0.404 |

### 4.5.6 Analysis on Model Complexity

As shown in Table 8, we compare the parameters of ALMT with other state-of-the-art Transformer-based methods. Due to the different hyper-parameter configurations for each dataset may lead to a slight difference in the number of parameters calculated. We calculated the model parameters under the hyper-parameter settings on the MOSI. Obviously, our ALMT obtains the best performance (Acc-7 of 49.42 %) with a second

computational cost (2.50M). It shows that ALMT achieves a better trade-off between accuracy and computational burden.

Table 8: Analysis on model complexity. Note: the parameter of other Transformer-based methods was calculated by authors from open source code with default hyper-parameters on MOSI.

| Method | Parameter | Acc-7 on MOSI |
|---|---|---|
| MuLT | 2.57M | 40.00 |
| MISA | 3.10M | 42.3 |
| MAG-BERT | **1.22M** | 43.62 |
| **ALMT** | 2.50M | **49.42** |

### 4.5.7 Visualization of Attention in AHL

In Figure 4, we present the average attention matrix (*i.e.,* $\alpha$ and $\beta$) on CH-SIMS. As shown, ALMT pays more attention to the visual modality, indicating that the visual modality provides more complementary information than the audio modality. In addition, from Table 3, compared to removing audio input, the performance of ALMT decreases more obviously when the video input is removed. It also demonstrates that visual modality may provide more complementary information.

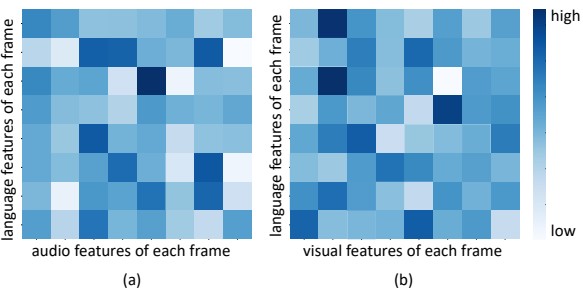

Figure 4: Visualization of average attention weights from the last AHL layer on CH-SIMS dataset. (a) Average attention matrix $\alpha$ between language and audio modalities; (b) average attention matrix $\beta$ between language and visual modalities. Note: darker colors indicate higher attention weights for learning.

### 4.5.8 Visualization of Robustness of AHL

To test the AHL's ability to perceive sentiment-irrelevant information, as shown in Figure 5, we visualize the attention weights ($\beta$) of the last AHL layer between language features ($H_l^3$) and visual features ($H_v^1$) on CH-SIMS. More specifically, we first randomly selected a sample from the test set. Then we added random noise to a peak frame

(marked by the black dashed boxes) of $H_v^1$, and finally observed the change of attention weights between $H_l^3$ and $H_v^1$. It is seen that when the random noise is added to the peak frame, the attention weights between language and the corresponding peak frame show a remarkable decrease. This phenomenon demonstrates that AHL can suppress sentiment-irrelevant information, thus obtaining a more robust hyper-modality representation for multimodal fusion.

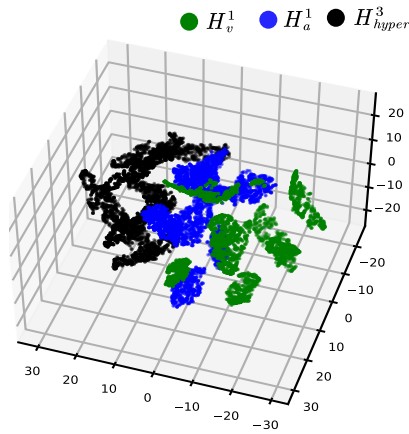

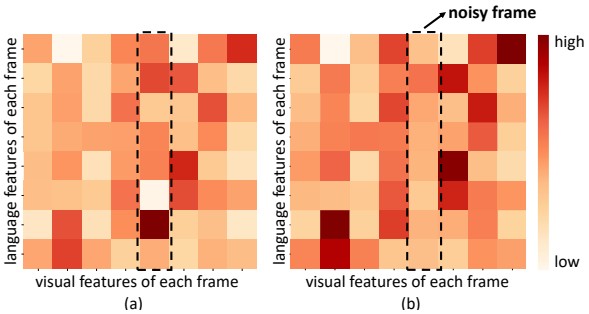

Figure 5: Visualization of the attention weights between language and visual modalities learned by the AHL for a randomly selected sample with and without a random noise on the CH-SIMS dataset. (a) The attention weights without a random noise; (b) the attention weights with a random noise. Note: darker colors indicate higher attention weights for learning.

### 4.5.9 Visualization of Different Representations

In Figure 6, we visualized the hyper-modality representation $H_{hyper}^3$, visual representation $H_1^v$ and audio representation $H_1^a$ in a 3D feature space by using t-SNE (Van der Maaten and Hinton, 2008) on CH-SIMS. Obviously, there is a modality distribution gap existing between audio and visual features, as well as within their respective modalities. However, the hyper-modality representations learned from audio and visual features converge in the same distribution, indicating that the AHL can narrow the difference of inter-/intra modality distribution of audio and visual representations, thus reducing the difficulty of multimodal fusion.

### 4.5.10 Visualization of Convergence Performance

In Figure 7, we compared the convergence behavior of ALMT with three state-of-the-art methods (*i.e.,* MulT, MISA and Self-MM) on CH-SIMS. We choose the MAE curve for comparison as MAE indicates the model's ability to predict fine-grained sentiment. Obviously, on the training set, although

Figure 6: Visualization of different representations in 3D space by using t-SNE.

Self-MM converges the fastest, its MAE of convergence is larger than ALMT at the end of the epoch. On the validation set, ALMT seems more stable compared to other methods, while the curves of other methods show relatively more dramatic fluctuations. It demonstrates that the ALMT is easier to train and has a better generalization capability.

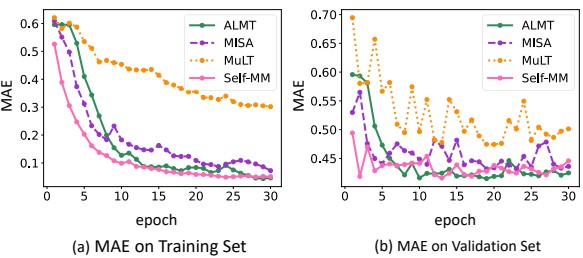

Figure 7: Visualization of convergence performance on the train and validation sets of CH-SIMS. (a) The comparison of MAE curves on the training set; (b) the comparison of MAE curves on the validation set. Note: the results of other methods reproduced by authors from open source code with default hyper-parameters.

## 5 Conclusion

In this paper, a novel Adaptive Language-guided Multimodal Transformer (ALMT) is proposed to better model sentiment cues for robust Multimodal Sentiment Analysis (MSA). Due to effectively suppressing the adverse effects of redundant information in visual and audio modalities, the proposed method achieved highly improved performance on several popular datasets. We further present rich in-depth studies investigating the reasons behind the effectiveness, which may potentially advise other researchers to better handle MSA-related tasks.

## Limitations

Our AMLT which is a Transformer-based model usually has a large number of parameters. It requires comprehensive training and thus can be subjected to the size of the training datasets. As current sentiment datasets are typically small in size, the performance of AMLT may be limited. For example, compared to classification metrics, such as Acc-7 and Acc-2, the more fine-grained regression metrics (*i.e.,* MAE and Corr) may need more data for training, resulting in relatively small improvements compared to other advanced methods.

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

## A Hyper-parameters

In this section, we show the selection of some key hyper-parameters on the validation set of CH-SIMS.

### A.1 Overview

We used PyTorch to implement our method. The experiments were conducted on a PC with Intel(R) Xeon(R) 6240C CPU at 2.6GHz and 128GB memory and NVIDIA GeForce RTX 3090. The key parameters are shown in Table 9. We see that most hyper-parameters are the same across these datasets, demonstrating the ALMT does not require complex hyper-parameters adjustment.

Table 9: Hyper-parameters of ALMT we use on the different datasets

|  | MOSI | MOSEI | CH-SIMS |
|---|---|---|---|
| Modality Feature Length $T$ | 8 | 8 | 8 |
| Vector Dimension $d$ | 128 | 128 | 128 |
| Modality Embedding Depth | 1 | 1 | 1 |
| AHL Depth | 3 | 3 | 3 |
| Fusion Transformer Depth | 2 | 4 | 4 |
| Batch Size | 64 | 64 | 64 |
| Initial Learning Rate | 1e-4 | 1e-4 | 1e-4 |
| Optimizer | AdamW | AdamW | AdamW |
| Epochs | 200 | 200 | 200 |
| Warm Up | ✓ | ✓ | ✓ |
| Cosine Annealing | ✓ | ✓ | ✓ |

### A.2 Effects of Length Settings of Modality Feature

In Figure 8, we show the effect of the sequence length $T$ of the token $H_m^0$ in modality embedding on the CH-SIMS dataset. It is observed that there are significant performance changes when the hyper-parameter is changed. And a similar phenomenon occurred on the MOSI and MOSEI datasets. Although the MAE is not the best when the Acc-5 is highest, *e.g.*, $T$ is set to 32. Considering that the model computation rises when the $T$ increases, we set $T$ to 8, which is beneficial for ALMT to obtain the best performance with a relatively lower computational cost.

### A.3 Effects of Depth Settings of AHL

Figure 9 presents the effect of AHL depth settings for MSA. Obviously, the ALMT achieves the best performance on the two most difficult evaluation metrics, *i.e.*, Acc-5 and MAE. Hence, in this study, we set the depth of AHL to 3. Moreover, on the MOSI and MOSEI datasets, we set it to 3 as the similar phenomenon is also observed.

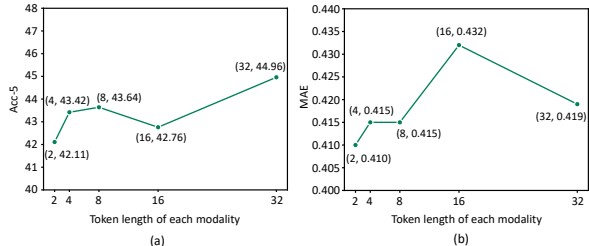

Figure 8: Effects of Token Length Settings in Modality Embedding. (a) Accuracy curve of different Token sequence lengths on the CH-SIMS; (b) MAE curve of different Token sequence lengths on the CH-SIMS

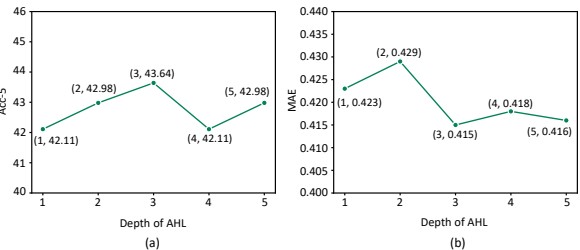

Figure 9: Effects of Depth Settings of AHL. (a) Accuracy curve of different AHL depth on the CH-SIMS; (b) MAE curve of different AHL depth on the CH-SIMS

### A.4 Effects of Depth Settings of Fusion Transformer

In Figure 10, we presents the effects of depth settings of cross-modality fusion Transformer on CH-SIMS. We observed that the ALMT can obtain the best result on Acc-5 and MAE when the depth is set to 3 and 5, respectively. However, to balance performance and model computation, we set the depth to 4 on the CH-SIMS. Following the similar rule, we set the depth of the cross-modality fusion Transformer to 2 and 4 on the MOSI and MOSEI, respectively.

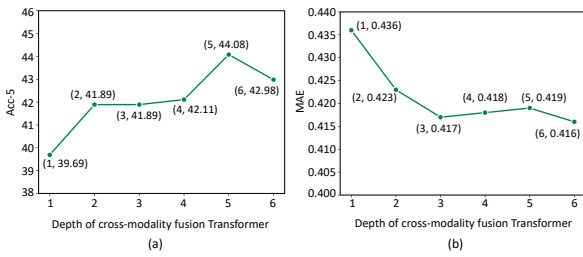

Figure 10: Effects of Depth Settings of Cross-modality Fusion Transformer. (a) Accuracy curve of different depth settings on the CH-SIMS; (b) MAE curve of different depth settings on the CH-SIMS