# OpenReview forum: "Learning Language-guided Adaptive Hyper-modality Representation for Multimodal Sentiment Analysis"
_EMNLP/2023/Conference — EMNLP 2023 Main_

### Official Review · Reviewer_PJLs · 2023-08-03

**Soundness:** 4

**Excitement:**

4: Strong: This paper deepens the understanding of some phenomenon or lowers the barriers to an existing research direction.

**Paper Topic And Main Contributions:**

This paper is about Adaptive Language-guided Multimodal Transformer, which incorporates an Adaptive Hyper-modality Learning module to learn an irrelevance/conflict-suppressing representation from visual and audio features under the guidance of language features at different scales for multimodal sentiment analysis task. The main contributions of the paper are: Proposed a novel multimodal sentiment analysis method, namely Adaptive Language-guided Multimodal Transformer, and explored a novel Adaptive Hyper-modality Learning module for representation learning.

**Reasons To Accept:**

1. The paper is written well.
2. Explored ALMT with AHL module which is quite interesting.
3. Ablation study is good.

**Reasons To Reject:**

1. The approach looks fine, but only one minor thing is it is good to share the results different fusion techniques as well.

**Reproducibility:**

4: Could mostly reproduce the results, but there may be some variation because of sample variance or minor variations in their interpretation of the protocol or method.

**Reviewer Confidence:**

3: Pretty sure, but there's a chance I missed something. Although I have a good feel for this area in general, I did not carefully check the paper's details, e.g., the math, experimental design, or novelty.

---

> ### Author Rebuttal · Authors · 2023-08-26
>
> # Response to Reviewer PJLs
>
> We sincerely thank you for your kind and constructive comments. Regarding the questions, we have the following answers.  We will try our best to incorporate the suggestions in the revised version of the paper.
>
> Regarding the comparison of different fusion techniques, we conducted some supplementary experiments, whose results are shown in the following table. Note that on the MOSI dataset, the use of Cross-modality Fusion Transformer to fuse language features and hyper-modality features is the most effective. On the CH-SIMS dataset, although TFN achieves better performance on the MAE metric, its Acc-5 is lower. Overall, using Transformer for feature fusion is an effective way.
>
> |Fusion Method|Acc-7 on MOSI|MAE on MOSI|Acc-5 on CH-SIMS|MAE on CH-SIMS|
> |----------------------|:----------------------:|:----------------------:|:----------------------:|:----------------------:|
> |Feature Concatenation|48.69|0.703|43.76|0.410|
> |Feature Addition|46.36|0.706|42.45|0.411|
> |LSTM|47.67|0.700|23.63|0.554|
> |BiLSTM|47.52|0.707|26.26|0.540|
> |GRU|47.81|0.710|44.86|0.414|
> |Tensor Fusion (TFN)|47.23|0.710|44.20|**0.403**|
> |Low-rank Multimodal Fusion (LMF)|46.65|0.715|45.08|0.408|
> |**Cross-modality Fusion Transformer**|**49.42**|**0.683**|**45.73**|0.404|

---

### Official Review · Reviewer_WX81 · 2023-08-05

**Soundness:** 4

**Excitement:**

4: Strong: This paper deepens the understanding of some phenomenon or lowers the barriers to an existing research direction.

**Paper Topic And Main Contributions:**

This paper presents a novel approach to Multimodal Sentiment Analysis (MSA) by introducing the Adaptive Language-guided Multimodal Transformer (ALMT). The primary problem addressed by this paper is the potential sentiment-irrelevant and conflicting information across different modalities (language, video, and audio) that may hinder the performance of MSA. The authors propose an Adaptive Hyper-modality Learning (AHL) module that learns an irrelevance/conflict-suppressing representation from visual and audio features under the guidance of language features at different scales. The paper claims that the proposed model achieves state-of-the-art performance on several popular datasets such as MOSI, MOSEI, and CH-SIMS.

**Questions For The Authors:**

Can the authors provide more details on how the Adaptive Hyper-modality Learning (AHL) module works, specifically how it suppresses sentiment-irrelevant information? Is there any theoretical justification for the chosen architecture, especially the design of AHL in the proposed method?

**Reasons To Accept:**

1) The paper presents a novel approach to tackle the issue of sentiment-irrelevant and conflicting information across different modalities in MSA, which is a significant contribution to the field.
2) The proposed model, ALMT, incorporates an Adaptive Hyper-modality Learning (AHL) module, which is a novel concept that could potentially inspire future research in this area.
3) The paper provides empirical evidence of the model's effectiveness by demonstrating state-of-the-art performance on several popular datasets.

**Reasons To Reject:**

It appears that the improvement of ALMT over the CHFN method on the MOSI dataset is quite marginal. It would be beneficial if the authors could conduct significance tests to validate the statistical significance of the observed improvements.

**Reproducibility:**

4: Could mostly reproduce the results, but there may be some variation because of sample variance or minor variations in their interpretation of the protocol or method.

**Reviewer Confidence:**

3: Pretty sure, but there's a chance I missed something. Although I have a good feel for this area in general, I did not carefully check the paper's details, e.g., the math, experimental design, or novelty.

---

> ### Author Rebuttal · Authors · 2023-08-26
>
> # Response to Reviewer WX81
> We are thankful to you for your time and effort in reviewing the paper. We also appreciate all the comments and suggestions.
>
> **Regarding the reasons to reject, we will clarify it in 4 points below.**
>
> 1. **Since CHFN and FDMER do not have open-source codes, we could not perform significance tests directly. If possible in the future, we will add the relevant results to the final version.** Moreover, we selected several SOTA methods (*e.g.,* MMIM and Self-MM) that have open-source code to conduct significance tests (Analysis of Variance, ANOVA) to validate the statistical significance of the observed improvements. The experimental result is shown in the following table. Obviously, our method is statistically more significant in terms of performance improvement because of the low p-value, which indicates a significant improvement in the prediction results of ALMT compared to other methods.
> ||p-value|
> |----------------------|----------------------|
> |ALMT & MMIM|0.027|
> |ALMT & Self-MM|0.116|
> 2. We emphasize that ALMT improves on almost all metrics in all datasets, while in some settings the improvement is quite significant. This serves to demonstrate the effectiveness and generalizability of our approach.
> 3. It can be seen that ALMT improves significantly on the Acc-7 and Acc-5 of MOSI, which also shows the effectiveness of our method. Regarding the reason why there is not much improvement on some metrics such as Acc-2, we believe that it is because these metrics are less difficult. After years of research, existing methods may tend to approach the performance boundary on these simple metrics.
> 4. The MOSI dataset is small in size (only 1284 samples in the training set), and we found that ALMT is always quickly overfitted during training. As well acknowledged, the Transformer-based model usually requires a large amount of data for training in order to realize its full performance. Therefore, we believe this is also one potential factor that limits the performance improvement of ALMT.
>
> **Regarding the question for the authors, the details about the AHL will be clarified in 4 points.**
>
> 1. As shown in Figure 3 of the paper, AHL uses the dominant modality (language) as Query, and the audio and visual modalities as Key and Value. In this way, AHL is able to query the audio and visual modalities for potentially sentiment-relevant and complementary information to the language, respectively, where the Transformer's multi-attention plays a crucial role. For useful information, the attention mechanism will give higher weight scores to highlight this information, while for redundant information (*e.g.,* noise), the attention mechanism will give lower weight scores to suppress this information.
> 2. In Section 4.5.6 of the paper, we visualize the AHL weights and it can be seen that on the CH-SIMS dataset, the model is guided by language with higher attention against the video information. It indicates that the visual modality provides more valid information than the audio modality on this dataset. This is further verified in the experiments in Section 4.5.1, where Acc-5 and MAE drop more drastically when removing the visual modality compared to removing the audio modality.
> 3. In Section C of the Appendix, we first randomly selected a sample and added noise, and then observed the changes in the weights computed by AHL. We see that for frames with added noise, the weights given by AHL are smaller, implying that AHL is able to suppress redundant information that is not related to sentiment or does not complement the language.
> 4. Regarding the choice of architecture, based on the phenomenon we observed and our motivations, we empirically believe that building ALMT/AHL based on Transformer is more appropriate. In addition, we have also done some experiments, and the experimental results show that Transformer-based architecture can better realize our idea and achieve better performance. For more details, please refer to Response to Reviewer JSNF and Reviewer PJLs.

---

### Official Review · Reviewer_JSNF · 2023-08-05

**Soundness:** 4

**Excitement:**

4: Strong: This paper deepens the understanding of some phenomenon or lowers the barriers to an existing research direction.

**Paper Topic And Main Contributions:**

This work proposes Adaptive Language-guided Multimodal Transformer (ALMT) to better model sentiment cues for robust Multimodal Sentiment Analysis (MSA). ALMT consists of three major components, i.e., modality embedding, adaptive hyper-modality learning, and multimodal fusion. Due to effectively suppressing the adverse effects of redundant information in visual and audio modalities, the proposed method achieved highly improved performance on several popular datasets. Detail experiments and analyses are provided to prove the effectiveness of the ALMT.

**Reasons To Accept:**

- This work presents a novel multimodal sentiment analysis method, namely Adaptive Language-guided Multimodal Transformer (ALMT), which for the first time explicitly tackles the adverse effects of redundant and conflicting information in auxiliary modalities (i.e., visual and audio modalities), achieving a more robust sentiment understanding performance.
- This work devises a novel Adaptive Hyper-modality Learning (AHL) module for representation learning. The AHL uses different scales of language features to guide the visual and audio modalities to form a hyper modality that complements the language modality.
- State-of-the-art performance and detailed analysis in several public and widely are provided.

**Reasons To Reject:**

- Novelty limitations. This is an incremental job. It is common to use Transformers to fuse multiple modalities.

**Reproducibility:**

4: Could mostly reproduce the results, but there may be some variation because of sample variance or minor variations in their interpretation of the protocol or method.

**Reviewer Confidence:**

4: Quite sure. I tried to check the important points carefully. It's unlikely, though conceivable, that I missed something that should affect my ratings.

---

> ### Author Rebuttal · Authors · 2023-08-26
>
> # Response to Reviewer JSNF
>
> We sincerely appreciate your careful and thoughtful comments. We have made every effort to address all the concerns raised. Below, we will discuss the issues.
>
> Due to the ability to efficiently model long time sequences and its multi-head attention mechanism, Transformer has demonstrated more powerful modeling capabilities than other methods (*e.g.,* LSTM and GRU) in many tasks of multimodal learning. However, it is worth noting that most of the previous studies in the MSA (*e.g.,* MISA and MulT) have used an equal perspective to treat each modality, directly fusing the features of each modality as inputs to Transformer.  We observed that the contribution of each modality to the recognition accuracy in MSA datasets is significantly different, and we believe that the above approach will limit the modeling ability of Transformer in MSA tasks. Therefore, we sought to design a dominant modality (language modality) guided sentiment representation learning model (ALMT) based on Transformer. **Although Transformer has been already very common in the field of multimodal learning, our novelty is in designing a dominant modality-guided learning paradigm for the first time. In experiments, aside from remarkable performance, we also provide rich analysis and evidence supporting the significance of the proposed method.**
>
> In addition, we found some interesting phenomena. We have tried to use different methods instead of AHL for representation learning. Experimental results are shown in the table below. We found that AHL based on Transformer design achieved the best performance, followed by MLP, while LSTM and GRU showed lower performance. If time and space permit, we will further analyze why LSTM and its variant networks perform so poorly in the final version.
>
> |Hyper-modality Learning  Method|Acc-7 on MOSI|MAE on MOSI|Acc-5 on CH-SIMS|MAE on CH-SIMS|
> |----------------------|:----------------------:|:----------------------:|:----------------------:|:----------------------:|
> |MLP (w/o language guidance)|34.40| 0.952       |38.29| 0.444          |
> |LSTM  (w/o language guidance)| 21.14         | 1.383       | 26.26            | 0.571          |
> |BiLSTM  (w/o language guidance)| 21.87 | 1.382       | 29.32            | 0.529          |
> |GRU  (w/o language guidance)| 22.16 | 1.377       | 28.67            | 0.529          |
> |**AHL**| **49.42**     | **0.683**   | **45.73**        | **0.404**      |

---

### Meta-Review · Area_Chair_8ScM · 2023-09-26

**Recommendation:** 4

**Metareview:**

All the reviewers rated the work as 4, across all categories.

Reviewers largely agree that:
-The paper is well written
-Experiments were interesting and novel
-The work achieves state-of-the-art performance

Some reviewers also noted that the gap between the prior state-of-the-art and the SotA that the authors achieve "seem[s] marginal".

---

### Decision · Program_Chairs · 2023-10-07

**Decision:**

Accept-Main

**Comment:**

All the reviewers rated the work as 4, across all categories.

Reviewers largely agree that:
-The paper is well written
-Experiments were interesting and novel
-The work achieves state-of-the-art performance

Some reviewers also noted that the gap between the prior state-of-the-art and the SotA that the authors achieve "seem[s] marginal".